# The Effects of High-Intensity Power Training versus Traditional Resistance Training on Exercise Performance

**DOI:** 10.3390/ijerph19159400

**Published:** 2022-07-31

**Authors:** Yu-Hua Chang, Yi-Chen Chou, Yun-Chi Chang, Kok-Hwa Tan, Mei-Hsuan Wu

**Affiliations:** 1Graduate Institute of Sport Coaching Science, Chinese Culture University, Taipei City 11114, Taiwan; yhchang@mx.nthu.edu.tw; 2Department of Athletic, National Tsing Hua University, Hsinchu City 30013, Taiwan; ycchou@mx.nthu.edu.tw; 3Department of Physical Education, Fu-Jen Catholic University, New Taipei City 242, Taiwan; james76630@hotmail.com; 4Physical Education Office, National Tsing Hua University, Hsinchu City 30013, Taiwan; khtan@mx.nthu.edu.tw; 5Precision Medicine Ph.D. Program, National Tsing Hua University, Hsinchu City 30013, Taiwan

**Keywords:** high intensity power training, resistance training, peak anaerobic power

## Abstract

Background: High-intensity interval training (HIIT) features short, repeated bursts of relatively vigorous exercise with intermittent periods of rest or low-intensity exercise. High-intensity power training (HIPT), in combination with HIIT and traditional resistance training (TRT), is characterized as multijoint high-intensity resistance exercises with low interset rest periods. HIPT requires people to finish the exercise as fast as possible, which increases acute physiological demands. The aim of the study was to investigate the differences between eight-week HIPT or TRT on exercise performance. Methods: Twenty-four college students were recruited and randomly assigned to either the HIPT or TRT group in a counterbalanced order. The power of upper and lower limbs (50% 1RM bench press and vertical jump) and anaerobic power were tested before and after the training (weeks 0 and 9). The results were analyzed by two-way analysis of variance (ANOVA) or Friedman’s test with a significance level of *α* = 0.05 to compare the effects of the intervention on exercise performance. Results: There were significant differences in the explosive force of the upper and lower limbs between the pretest and post-test in both the HIPT and TRT groups (*p* < 0.05). However, only the HIPT group showed a significant difference in the mean power on the Wingate anaerobic test between the pretest and post-test (*p* < 0.05). Conclusions: Both HIPT and TRT can improve upper and lower limb explosive force. HIPT is an efficient training protocol, which took less time and produced a better improvement in mean anaerobic power.

## 1. Introduction

Traditional resistance training (TRT) is used to improve maximum power through various multijoint resistance training exercises, which are usually performed two to three times per week, using few repetitions with the optimal load together with high interset rest periods. Typical TRT contains maximal dynamic power training (i.e., bench press and squat) and power training such as the power clean, power snatch, and weighted countermovement jump [1,2,3]. Long-term TRT can significantly improve the performance of sprinting, endurance running, or endurance cycling [4]. Resistance training can improve muscle adaptability, which is controlled by different factors such as the amount of training, training intensity, resting time, or training frequency [5]. In particular, muscle power and blood lactate are sensitive to resting time. If the resting time is less than one minute, lactic acid accumulates, and the synthesis of phosphocreatine is insufficient, which might result in decreased muscle output [6]. Different muscle training goals should match resting times between sets. To increase muscle strength, the resting time should be at least 3 min for every repetition. For the goal of muscle hypertrophy, the resting time should be less than the time of the total recovery of strength, which is 30 to 60 s. In terms of improving muscle endurance, it is suitable to adopt cyclic resistance training that features a shorter resting period, such as 30 s [7]. To summarize, the resting time should be adjusted according to the different training goals.

In the past few years, high-intensity interval training (HIIT) has replaced TRT in reinforcing aerobic fitness. HIIT features short, repeated bursts of relatively high intensity exercises with a short inter-rest period or a low-intensity recovery activity [8,9]. HIIT is useful for many people because it takes much less time compared with TRT. HIIT combined with multijoint high-intensity resistance training is called high-intensity power training (HIPT). Compared with HIIT, HIPT involves shorter and irregular resting intervals and continuous multijoint high-intensity exercise as positive stimulations of maximum aerobic capacity and muscular strength [3]. In addition, HIPT is often coupled with gymnastics such as parallel bars, hand rings, and hand-standing [9]. 

In HIPT, little time is required to reach high training intensity compared with TRT, and it effectively improves power performance and maximum aerobic speed performance [3,10]. In terms of training mode, the differences between HIPT and TRT are the resting interval and the order of training. The resting interval is around 90 s between sets in TRT, while the HIPT is carried out in cycle: the resting interval is 15 s between each movement with three to five cycles in total. The training time of TRT is four to five times longer than that of HIPT [3]. Although there are systematic review articles that summarize the effects of HIPT in terms of aerobic fitness, muscular fitness, or psycho-physiological parameters, it was hard to draw a definite conclusion on the differences between HIPT and other concurrent training methods due to the disparate styles of HIPT and the heterogeneous intervention protocols and outcome measurements [9,11,12]. More studies are required to establish a training prescription guideline for HIPT.

To summarize, HIPT takes less time and improves aerobic fitness adaptations and power performance, while TRT takes longer time, but it elevates the sprint speed and endurance performance. Accordingly, the aim of this study was to investigate the differences between HIPT and TRT in terms of the explosiveness of the upper and lower limbs and anaerobic power.

## 2. Materials and Methods

### 2.1. Subjects

We recruited healthy college students with no psychological or physiological disorders as participants, who were randomly assigned to either HIPT (N = 12) or TRT (N = 12) groups (17 men, aged 21.47 ± 1.8 years). Before the study, subjects were informed of the purpose, process, and precautions of the study. All subjects provided the informed consent form. The study was approved by the Research Ethics Committee of the university.

### 2.2. Experimental Design

This study was designed as a randomized parallel trial. After the subjects signed the informed consent, they were instructed in the movements of the training protocol (i.e., bench press, squat, upright barbell row, and heel raise) and the training assessment (i.e., bench press, vertical jump, and the Wingate test) to ensure their safety. Before the intervention, the subjects were required to test their maximum muscle strength in each movement in the training protocol on the appointed day. Subjects participated in the interventional training protocol for 8 weeks with a frequency of three times per week for a total of 24 training sessions. There was at least a 24 h rest between each training session to ensure a complete recovery. The subjects also needed to complete assessments before and after the 8-week HIPT or TRT training protocol.

#### 2.2.1. HIPT Protocol

The participants in the HIPT group were required to perform 3 rounds of training with 60% of their maximum muscle strength (i.e., 60% of one-repetition maximum (1RM)). The participants were required to perform bench presses, squats, upright barbell rows, and heel raises. Each movement was performed for 10 s, and the break between exercises was 15 s, which allowed participants to move safely between exercises. There was a 90 s interval between rounds. The intervention took around 8 min in total for each training session (Figure 1).

#### 2.2.2. TRT Protocol

The participants in the TRT group were required to perform 3 sets of training with 90% of 1RM in the order of bench presses, squats, upright barbell rows, and heel raises. Each movement was performed 3 times and the rest between exercises was 180 s. The intervention took around 40 min for each training session (Figure 2).

### 2.3. Assessment

All subjects underwent a pretest at week 0 and a post-test at week 9. Before each test, the subject would perform the same warm-up: jogging for 8 min, bounding drill with cone 5 times, mark drills (i.e., mark A, B, C, D, half- and high-lifting running) and 50-meter strides. The tests included upper and lower limb explosive force and anaerobic power.

#### 2.3.1. Upper Limb Explosive Force (Bench Press)

The upper limb explosive force was measured with an accelerometer (MetaMotionR, MbientLab Inc., San Francisco, CA, USA) when conducting 50% of the 1RM bench press. The sensor was fixed to the center of the barbell. The acceleration data were collected by a Bluetooth connection via the MetaWear app. After the starting signal, subjects were asked to finish the bench presses as quickly as they could. The instructor gave the starting signal 3 times, and each subject’s maximum acceleration were recorded and averaged. The bench press force (F) was calculated by F = m × a, where m refers to the total mass of the barbells and weights, and a is the acceleration of the y-axis measured by the accelerometer.

#### 2.3.2. Lower Limb Explosive Force (Vertical Jump)

We measured the height of jumps with a measuring tape. Before the test began, the subjects stood on a 100 cm square with their feet shoulder-width apart. The subjects then bent their knees and jumped upward with both arms swinging upward. They tried to keep their body in a straight line while rising into the air. They touched a board when they reached the highest point of their jump. Finally, the subjects slightly bent their knees when landing. Each subject conducted vertical jump tests twice, and the highest jump was recorded.

#### 2.3.3. Anaerobic Power Test (Wingate Anaerobic Test)

The aim of the Wingate test is to measure the anaerobic power of the lower body. The peak power outputs and the mean power outputs were measured on a bicycle’s ergometer (MedGraphics Corival Cycle Ergometer, the Netherlands). Subjects performed a cycling warm-up with no resistance until 60 rpm was reached, and they maintained speed. When the starting signal was given, subjects were instructed to pedal as fast as possible for 30 s with a predetermined resistance load of 75 g/kg body weight (i.e., personal weight × 0.075 kg). During the 30 s test, the instructor kept verbally encouraging subjects to pedal at maximum speed.

### 2.4. Statistical Analysis

The data were analyzed using SPSS 20.0 software (IBM Corp, Armonk, NY, USA). Descriptive statistics are presented with the medians, lower and upper quartiles (Q1, Q3), means (M), and standard deviations (SD). The Shapiro–Wilk test was used to test whether the data followed a normal distribution. The difference in the indices between the HIPT and the TRT groups was tested by the Mann–Whitney U test. The differences between the upper and lower limb explosive forces and anaerobic power were tested according to the normal distribution. Data were analyzed using a two-way repeated analysis of variance (ANOVA) with the group as a between-subjects variable and with the repeated measures as the time factor (within-subjects: pretest and post-test). Otherwise, the Friedman test was applied to test the differences between the pretest and the post-test among groups if the data followed non-normal distribution. The significant level was set at *α* = 0.05.

## 3. Results

Twenty-four healthy college students were randomly assigned to either the TRT or HIPT group. Three of the subjects in the TRT group did not finish the training program, resulting in 12 and 9 subjects in the HIPT and TRT groups, respectively. The variables were first tested with the normal distribution (Appendix A). The baseline information of the two groups is shown in Table 1.

### 3.1. Differences in Upper Limb Explosive Force between Pretest and Post-test

In the pretest, the bench press (50% of 1RM) scores were analyzed by the Mann–Whitney U test to compare the differences in the explosive force of the upper limbs between the HIPT and the TRT groups. There were no significant differences between the power of the upper limbs in the pretest. The results showed that the medians of the bench press for the HIPT group in the pretest and post-test were 162.01 and 190.00 N, respectively. As for the TRT group, the medians of the bench press were 172.74 and 203.11 N before and after the training intervention, respectively. The bench press outcomes of the HIPT and TRT groups in the post-test were both better than in the pretest, and the differences were both significant (*p* < 0.05) (Table 2 and Table 3; Figure 3A).

### 3.2. Differences in Vertical Jump between Pretest and Post-test

The results showed that the average of the vertical jump of the HIPT group in the pretest and post-test was 57.00 ± 6.18 cm and 63.25 ± 5.58 cm, respectively; those of the TRT group were 57.11 ± 10.96 cm and 59.11 ± 10.69 cm, respectively (Table 2; Figure 3B). The two-way repeated ANOVA revealed a significant interaction between the groups and time (F (1,19) =14.677, *p* = 0.001; Table 4). The vertical jump outcomes of both the HIPT and TRT groups in the post-test were significantly better than those in the pretest (*p* < 0.001 and *p* = 0.001) (Figure 3B).

### 3.3. Differences in Anaerobic Power between Pretest and Post-Test

The results showed that the averages of the peak power output of the HIPT group in the pretest and post-test were 788.25 ± 156.23 W and 835.31 ± 122.74 W, respectively; those of the TRT group were 773.38 ± 148.66 W and 773.68 ± 140.18 W, respectively (Table 2; Figure 3C). The peak power outputs per kilogram of the HIPT group in the pretest and post-test were 11.37 ± 1.23 W·kg^−1^ and 12.10 ± 0.52 W·kg^−1^, respectively, while those of the TRT group were 11.68 ± 1.64 W·kg^−1^ and 11.66 ± 1.26 W·kg^−1^ respectively (Table 2). The two-way repeated ANOVA of peak power and peak power per kilogram both revealed neither significant interaction nor significant main effect between the groups and time (*p* > 0.05; Table 4).

In the pretest, there were no significant differences in the mean power output in the pretest between the HIPT and TRT groups by the Mann–Whitney U test (medians of HIPT: 642.37 W versus TRT: 676.49 W; *p* > 0.05; Table 2). The results showed that the medians of the mean power output of the HIPT group in the pretest and post-test were 642.37 and 746.69 W, respectively, while those of the TRT group were 676.49 and 696.10 W, respectively. The mean power outputs per kilogram of the HIPT group in the pretest and post-test were 9.44 and 10.70 W·kg^−1^, respectively, and those of the TRT group were 9.12 and 9.56 W·kg^−1^, respectively. The mean power outputs of the HIPT and the TRT groups in the post-test were both better than in the pretest, but only the HIPT group showed significant differences in the Friedman test (*p* = 0.001) (Table 2 and Table 3; Figure 3D). The mean power output per kilogram yielded a similar result on the Friedman test (*p* = 0.001) (Table 2 and Table 3).

## 4. Discussion

The aim of this study was to investigate the differences between eight-week TRT and HIPT on the explosive force of the upper and lower limbs and anaerobic power. Previous studies have proven that resistance exercise can significantly increase muscle strength, local muscle endurance, and anaerobic ability [3,4,13,14,15,16]. The variables that modulate training include (1) muscle movements, (2) load and quantity, (3) movement selection and order, (4) resting time, (5) repeated speed, and (6) frequency. These determine the adaptation of neuromuscular, neuroendocrine, and musculoskeletal systems to acute and chronic resistance exercises [5]. The present study adopted cyclic order, maximal speed, and 15 s resting times. The total training time of HIPT lasted only 8 min compared with the 40 min of TRT; in the former, the repetitions could be gradually increased. That is, subjects were able to execute at a better speed as their ability improved. The results showed a significant difference between the HIPT and the TRT groups in the explosive force of the upper and the lower limbs. However, only the mean anaerobic power improved in the HIPT group in the post-test. Accordingly, the results indicated that short HIPT sessions had a positive training effect. Alcaraz, Sánchez-lorente and Blazevich also pointed out that circuit training is a good way to reduce the time required for resistance training while meeting a sufficient training volume [17].

### 4.1. Explosive Force of Upper Limb

The explosive force of muscles is important for athletes because maximum strength is often required in sports. The maximum muscle strength is affected by a variety of neuromuscular factors, including fascicle length, muscle fiber composition, cross-sectional muscle area, pennation angle, and tendon compliance [18]. Muscle strength can be increased through resistance training, while high-intensity, rather than low-intensity, training is more effective for the development of maximum strength [19]. In this experiment, the TRT group performed bench press training with 90% of 1RM, and the HIPT group performed the same with 60% of 1RM as quickly as possible. We found that after 8 weeks of training, both groups were significantly different in terms of bench press. Alcaraz et al. showed that 8 weeks of high-resistance circulation training could significantly improve the performance of bench presses, and this study presented similar results [20]. Padulo et al. compared different speeds of bench press (i.e., 80–100% maximal speed and self-selected pushing speed) and of strength training at 85% of 1RM. After 3 weeks of training, the results revealed high-speed exercise could increase muscle strength by 10.20% [21]. Pareja-Blanco et al. found that repeating resistance training at the maximum velocity, compared with half-maximal concentric velocity, provided a superior stimulus for inducing neuromuscular adaptations, thus improving athletic performance [22]. Accordingly, in addition to intensity, the speed of movement was a factor that influenced the training effect.

### 4.2. Explosive Force of Lower Limb

It has been proven that strength training can significantly improve athletes’ vertical jump performance [23,24]. In the present experiment, the vertical jump performance of the HIPT and TRT groups both significantly improved. Izquierdo et al. compared bench presses and squats with different loads (60%, 65%, 70%, and 75% of 1RM) [25]. It was observed that training with a lower load and at a faster speed allowed for more repetitions to be executed in a fixed time. Therefore, as the muscle strength progresses, the HIPT group can achieve more repetitions during a training time, leading to an increase in the training volume. In addition, during exercise, muscle strength depends not only on nerve conduction and muscle excitability, but also on muscle contraction dynamics [26]. Explosive power (P) is the product of muscle force (F) and speed (V). To achieve optimal performance, muscle strength or speed can be improved to enhance explosive power [27]. Although resistance training can increase athletes’ muscle strength, it is recommended for experienced athletes to develop strength through a ballistic training approach [18]. Cormie et al. showed that resistance training increases maximal nerve activation and muscle thickness, while ballistic power training can increase the rate of electromyography (EMG) rise during 0–30% of 1RM jump squats, and that the changes in neural activation induced by training depended on the specific stimulus applied during the training [28]. If the rapid movement of HIPT was adopted, it could raise the muscle contraction speed and increase the output power [20]. The performance of vertical jumps depends on the maximum strength, strength development speed, muscle coordination, and stretch/short cycle, all of which are factors that can effectively increase explosive power [29].

### 4.3. Anaerobic Power

Strength training was reported to improve cyclists’ peak power output and mean power output on a 30 s Wingate test, as well as the explosive power with maximum speed [16,30]. In our study, the mean power output only improved in the HIPT group after 8nweeks of training. However, though the peak power output improved in the HIPT group, there was no significant difference. Romero-Arenas et al. randomly distributed 29 healthy men into three groups: traditional power training (n = 10), high-intensity power training (n = 10), and the control group (n = 9). These participants carried out 10 weeks of training three times per week. The results showed a significant increase in both mean power and peak power in the Wingate test. The training protocol in Romero-Arenas et al.’s study contained the Wingate test in traditional power training and high-intensity power training, which might explain the significant differences in peak power. These improvements on the Wingate test suggested an impact of this training model on anaerobic metabolism, which might result from neuromuscular adaptations such as higher levels of muscular phosphocreatine and increased activity of the anaerobic enzymes [3]. Tabata et al. found that high-intensity intermittent exhaustive exercise could improve VO_2max_ and increase anaerobic capacity by 28% [31]. Moreover, to improve anaerobic capacity, lactic acid production during training should be a focus. Márquez et al. showed that the effects of HIPT on muscle and metabolic reactions were stronger than TRT [10]. The HIPT group in this study produced a large amount of lactic acid, which resulted in muscle adaptation. This might explain why the HIPT group was more effective than the TRT group in improving mean power.

Our results indicated HIPT is an efficient training protocol, which is efficacious for improving explosive force and anaerobic power. Moreover, the COVID-19 pandemic has dramatically altered the lifestyle of human beings, including the training plans of athletes. Social distancing and confinement measures inevitably cause a reduction in general physical activity and an increase in sedentary time, which might indirectly influence mental health [32,33]. It is necessary and urgent to develop an efficient physical activity program to maintain physical and mental health. Accordingly, HIPT, along with HIIT, should be a physical activity adapted to the pandemic period to promote and maintain physical health, not only for healthy adolescents but also for those with chronic diseases such as diabetes [34,35].

This is a preliminary investigation comparing the effect of HIPT and TRT training protocols on the explosive force of upper and lower limbs and anaerobic power. However, there were some limitations to be considered. First, the number of subjects was small, so a larger population is needed to verify the outcome. The second limitation was that we did not take the subjects’ exercise habits into consideration. The college students might have taken different physical education classes during the 8-week training intervention. 

In addition, it is not clear whether a better training outcome can be achieved when strength training and endurance training are combined. Some studies suggested that the compatibility between high-intensity strength and endurance training might be greater [36]. However, other studies pointed out that endurance and strength training triggered different neurological adaptations. If both were simultaneously performed, maximum strength performance might be reduced [37]. Therefore, the relevant training modes extended by HIPT need to be investigated in future research.

## 5. Conclusions

HIPT effectively improved the explosive power of upper and lower limbs as well as anaerobic power in a shorter training time, so is suitable for people who have little time to exercise or as an adapted physical activity during the COVID-19 pandemic period. However, compared with TRT, the speed of performing HIPT training is faster, and the rest time is lower. It is recommended that people with regular exercise habits to conduct HIPT training or beginners under the guidance of a coach with relevant sports expertise.

## Figures and Tables

**Figure 1 ijerph-19-09400-f001:**
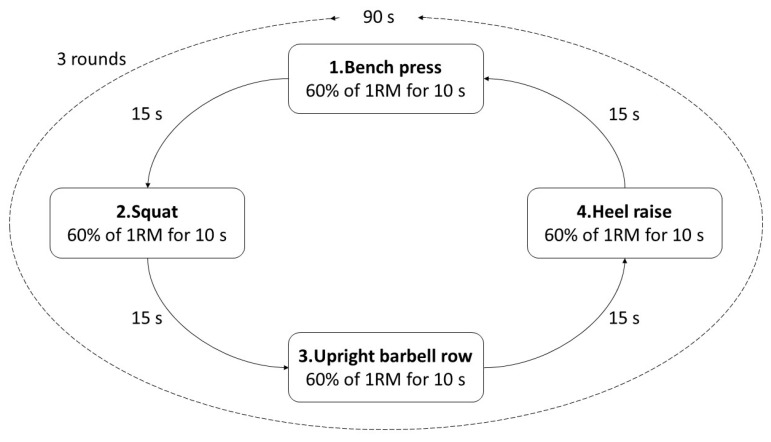
HIPT protocol.

**Figure 2 ijerph-19-09400-f002:**
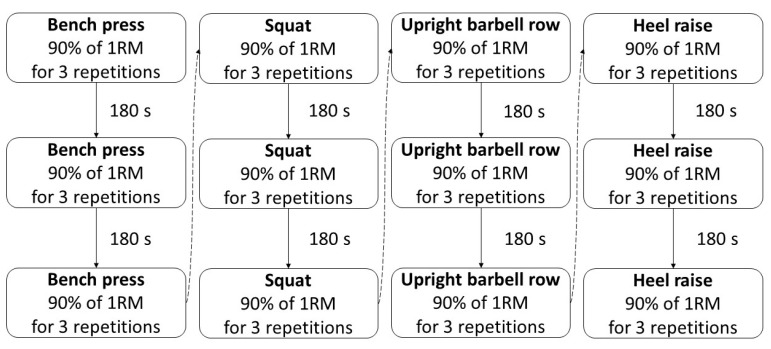
TRT protocol.

**Figure 3 ijerph-19-09400-f003:**
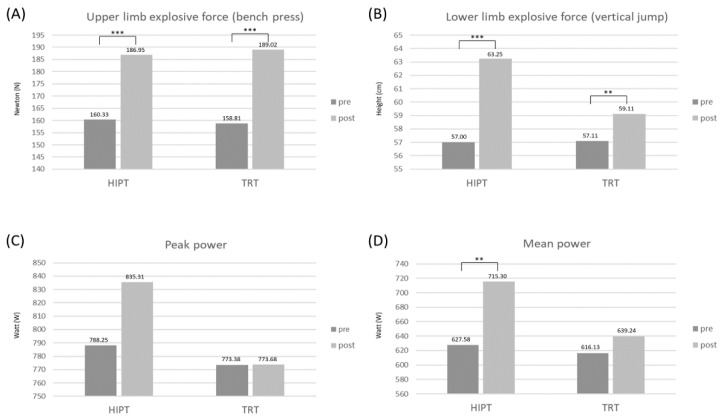
The results of pretest and post-test (after 8 weeks of training) of HIPT and TRT groups. (**A**) Bench press, (**B**) vertical jump, (**C**) peak power output in Wingate anaerobic test, and (**D**) mean power output in Wingate anaerobic test. ** *p* < 0.01, *** *p* < 0.001.

**Table 1 ijerph-19-09400-t001:** Baseline characteristics of the HIPT and TRT groups.

Groups	HIPT (N = 12)			TRT (N = 9)			*p*-Value
	Median	Q1	Q3	Median	Q1	Q3	
Male/Female	9/3			7/2			
Age (years)	21.00	20.00	22.75	22.00	19.50	24.00	0.469
Height (cm)	176.00	167.75	181.50	175.00	172.50	179.00	0.943
Weight (kg)	73.50	62.25	75.00	67.00	59.50	72.00	0.270
BMI (kg/m^2^)	22.44	21.73	24.27	21.79	20.38	23.92	0.255

The *p*-values were tested by Mann–Whitney U test.

**Table 2 ijerph-19-09400-t002:** Pretest and post-test of the HIPT and TRT groups.

Assessments	Group	Pre		Post	
		Mean ± SD	Median [Q1, Q3]	Mean ± SD	Median [Q1, Q3]
bench press (N)	HIPT	160.33 ± 47.72	162.01 [114.95, 188.31]	186.95 ± 43.72	190.00 [138.23, 236.29]
	TRT	158.81 ± 49.33	172.74 [113.35, 194.60]	189.02 ± 54.96	203.11 [148.63, 217.08]
vertical jump (cm)	HIPT	57.00 ± 6.18	59.00 [50.25, 62.75]	63.25 ± 5.58	65.50 [57.25, 68.00]
	TRT	57.11 ± 10.96	59.00 [47.50, 67.00]	59.11 ± 10.69	59.00 [50.00, 68.50]
peak power (W)	HIPT	788.25 ± 156.23	790.61 [663.55, 934.73]	835.31 ± 122.74	889.43 [718.84, 934.73]
	TRT	773.38 ± 148.66	825.55 [621.78, 900.03]	773.68 ± 140.18	823.55 [648.25, 895.32]
mean power (W)	HIPT	627.58 ± 98.64	642.37 [532.22, 711.34]	715.30 ± 104.74	746.69 [617.47, 798.46]
	TRT	616.13 ± 127.56	676.49 [468.25, 726.49]	639.24 ± 117.11	696.10 [526.78, 730.71]
peak power per kg (W·kg^−1^)	HIPT	11.37 ± 1.23	10.64 [10.54, 12.35]	12.10 ± 0.52	12.25 [11.48, 12.41]
	TRT	11.68 ± 1.64	11.31 [10.20, 13.14]	11.66 ± 1.26	11.47 [10.62, 12.70]
mean power per kg (W·kg^−1^)	HIPT	9.14 ± 1.09	9.44 [8.60, 9.77]	10.44 ± 1.24	10.70 [9.97, 11.20]
	TRT	9.29 ± 1.42	9.12 [7.74, 10.17]	9.64 ± 1.11	9.56 [8.63, 10.58]

**Table 3 ijerph-19-09400-t003:** The differences in bench press and mean power between pretest and post-test.

Group	Assessments	Chi-Square	df	*p*-Value
bench press (N)	HIPT	5.333	1	0.021 *
	TRT	9.000	1	0.003 **
mean power (W)	HIPT	12.000	1	0.001 **
	TRT	2.778	1	0.096
mean power per kg (W·kg^−1^)	HIPT	12.000	1	0.001 **
	TRT	2.778	1	0.096

* *p* < 0.05, ** *p* < 0.01 by Friedman test.

**Table 4 ijerph-19-09400-t004:** Two-way ANOVA of vertical jump and peak power.

Assessments	Effect	F	df	*p*-Value	Partial Eta Squared
vertical jump (cm)	group	0.275	1	0.606	0.014
	time (pre-post)	55.307	1	<0.001 ***	0.744
	Group × time	14.677	1	0.001 **	0.436
peak power (W)	group	0.351	1	0.560	0.018
	time (pre-post)	3.069	1	0.096	0.139
	Group × time	2.992	1	0.100	0.136
peak power per kg (W·kg^−1^)	group	0.014	1	0.906	0.001
	time (pre-post)	3.283	1	0.086	0.147
	Group × time	3.554	1	0.075	0.158

** *p* < 0.01, *** *p* < 0.001 by two-way repeated measure ANOVA.

## Data Availability

Not applicable.

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
