# Peer review of "The Effects of High-Intensity Power Training versus Traditional Resistance Training on Exercise Performance"

_ijerph, 2022, doi:10.3390/ijerph19159400_

Round 1
Reviewer 1 Report
Thanks for the consideration to review this work. The novelty of the topic and the study design is on point for the current exercise science research.
However, there are some slight issues that will improve the quality of the paper
Ln 61 - 77 This looks more like a discussion ratter than an introduction. Please mention the gaps that you are trying to address and/or solve with your study.
Ln 120 What brand/make the accelerometer is? How did you recorded? With and specialized app? software? analog to manually?
Ln 141 - 147 Stats overall
I will sujest to adjust the statistical analysis as a two-way repeated measures ANOVA [Time (pre vs post) vs Group (HIPT vs TRT]. Using t-test will infringe some variability issues in the raw data of your variables.
Ln 155 - 164 How do you calculate/record the Newton force from the upper limb explosive force?
Table 2. Please add the units among variables
Figures. I would sugest to re-do the figures to show individual data among groups x time
Results can be re-do after the 2x2 ANOVA is applied, along with not much modifications from discussion/conclusion
Author Response
Dear reviewer,
Thank you for your careful review. In the revised manuscript, we modified all the statistics method according to the reviews’ comments. The outcomes are somewhat different from the original version, so we also revised our discussion. The second one is we have revised our introduction—removed the retracted reference and add new references to make the aim of the study clearer. The third one is the methodology of the assessment of upper limb explosive force (bench press), which we clarify the use of accelerometer
The followings are the reply to the reviewers’ comments point by point.
Ln 61 - 77 This looks more like a discussion rather than an introduction. Please mention the gaps that you are trying to address and/or solve with your study.
ANS. Thank you for your suggestion. We have revised the paragraph to clarify the aim of the study.
Ln 120 What brand/make the accelerometer is? How did you recorded? With and specialized app? software? analog to manually?
ANS. The upper limb explosive force is measured with accelerometer (MetaMotionR, MbientLab) when conducting 50% of 1RM bench press. The sensor is fixed to the center of barbell. The acceleration data were collected by Bluetooth connection via MetaWear APP (MbientLab).
Ln 141 - 147 Stats overall
I will suggest to adjust the statistical analysis as a two-way repeated measures ANOVA [Time (pre vs post) vs Group (HIPT vs TRT]. Using t-test will infringe some variability issues in the raw data of your variables.
ANS. Thank you very much for your kindly suggestion. The reviewer2 also raised statistical suggestions. The sample size in our study is limited. So, we first tested the normal distribution of the continuous data (please refer to supplementary table 1). If the distribution was non-normal, we used non-parametric methods of statistics (i.e. Friedman test). If the distribution was normal, we revised the statistical analysis as a two-way repeated measures ANOVA.
Ln 155 - 164 How do you calculate/record the Newton force from the upper limb explosive force?
ANS. The bench press force (F) was calculated by F=m×a, where “m” referred to total mass of the barbell and weights, “a” referred to acceleration of y-axis from the accelerometer. We have added the description to 2.3.1 to make it clear.
Table 2. Please add the units among variables
ANS. Thank you for your reminder. We have added the units for each variable.
Figures. I would suggest to re-do the figures to show individual data among groups x time
ANS. Thank you for your advice. We have added the data to the figures and combined figures into one figure in order to make the paper easier to read.
Results can be re-do after the 2x2 ANOVA is applied, along with not much modifications from discussion/conclusion
ANS. Thank you for your kindly suggestion. We have revised the results and discussion when the new statistical analyses were performed.
Reviewer 2 Report
In the manuscript The Effects of High Intensity Power Training Versus Traditional Resistance Training on Exercise Performance, the authors review high intensity power training versus traditional strength training. The authors studied the effect of these two training programs on three indicators: explosive force of upper and lower limb, and the results of the Wingate anaerobic test and received some new facts. However, when reviewing the article, I had a number of comments.
1. The authors took the training program as a basis and cited a study several times in the manuscript Smith et al (2013). However, this article was withdrawn by the journal in 2017. At the same time, the following was written:
Retraction.
The Journal of Strength and Conditioning Research has retracted the article entitled “CrossFit-based High Intensity Power Training Improves Maximal Aerobic Fitness and Body Composition” by Smith, MM, Sommer, AJ, Starkoff, BE, and Devor, ST, published in the November 2013 issue of the Journal of Strength and Conditioning Research (Vol. 27, pp. 3159 - 3172). The Journal of Strength and Conditioning Research was advised on May 4, 2017 by one of the authors that the study was not conducted under an IRB approved protocol for that study, as was stated in the article. Because the study was performed without proper IRB approval, the article has been retracted. This retraction follows the October 2015 erratum which stated that “after the article was published, 10 of the 11 participants who did not complete the study have provided their reasons for not finishing, with only 2 mentioning injury or health conditions that prevented them from completing follow-up testing.” The injury and health conditions that prevented the participants from completing follow-up testing were not caused by participation in the CrossFit-related study, and a federal judge ruled in September of 2016 that the injury data reported by the authors in the article was false.
It is clear that after that this article cannot be cited. For example, the authors of the meta-analysis excluded this study from their analysis (Claudino JG, 2018).
2. The groups in the study were small, therefore, in statistical analysis, it is mandatory to check the data for normal distribution. When the distribution is different from normal, it is incorrect to use parametric methods of statistics (Student's t-test). Therefore, the conclusions of the study may be incorrect.
3. The obtained data are repeated in full in the tables, the text of the manuscript and in the figures. It's redundant. You can recommend the authors to remove the duplication of data, and try to conduct additional data analysis (correlations, add calculated indicators - per kg of body for example)
4. Table 2 is poorly designed, in particular, there are no units for measuring indicators.
5. The performance of the Wingate anaerobic test after the High Intensity Power Training program was previously studied by Romero-Arenas S, et al. For some reason, the name of the first author was shortened in the manuscript (Arenas), which is incorrect. It was also necessary to refer to this work (the authors mention the maximum aerobic test in this manuscript, but not about the Wingate anaerobic test) and compare their results with those obtained earlier.
References
1. CrossFit-based High Intensity Power Training Improves Maximal Aerobic Fitness and Body Composition: Retraction. J Strength Cond Res. 2017 Jul;31(7):e76. doi: 10.1519/JSC.0000000000001990.
2. Claudino JG, Gabbett TJ, Bourgeois F, Souza HS, Miranda RC, Mezêncio B, Soncin R, Cardoso Filho CA, Bottaro M, Hernandez AJ, Amadio AC, Serrão JC. CrossFit Overview: Systematic Review and Meta-analysis. Sports Med Open. 2018 Feb 26;4(1):11. doi: 10.1186/s40798-018-0124-5.
3. Romero-Arenas S, Ruiz R, Vera-Ibáñez A, Colomer-Poveda D, Guadalupe-Grau A, Márquez G. Neuromuscular and Cardiovascular Adaptations in Response to High-Intensity Interval Power Training. J Strength Cond Res. 2018 Jan;32(1):130-138. doi: 10.1519/JSC.0000000000001778.
Author Response
Dear reviewers,
Thank you for your careful review. In the revised manuscript, we modified all the statistics method according to the reviews’ comments. The outcomes are somewhat different from the original version, so we also revised our discussion. The second one is we have revised our introduction—removed the retracted reference and add new references to make the aim of the study clearer. The third one is the methodology of the assessment of upper limb explosive force (bench press), which we clarify the use of accelerometer
The followings are the reply to the reviewers’ comments point by point.
1.The authors took the training program as a basis and cited a study several times in the manuscript Smith et al (2013). However, this article was withdrawn by the journal in 2017. At the same time, the following was written:
Retraction.
The Journal of Strength and Conditioning Research has retracted the article entitled “CrossFit-based High Intensity Power Training Improves Maximal Aerobic Fitness and Body Composition” by Smith, MM, Sommer, AJ, Starkoff, BE, and Devor, ST, published in the November 2013 issue of the Journal of Strength and Conditioning Research (Vol. 27, pp. 3159 - 3172). The Journal of Strength and Conditioning Research was advised on May 4, 2017 by one of the authors that the study was not conducted under an IRB approved protocol for that study, as was stated in the article. Because the study was performed without proper IRB approval, the article has been retracted. This retraction follows the October 2015 erratum which stated that “after the article was published, 10 of the 11 participants who did not complete the study have provided their reasons for not finishing, with only 2 mentioning injury or health conditions that prevented them from completing follow-up testing.” The injury and health conditions that prevented the participants from completing follow-up testing were not caused by participation in the CrossFit-related study, and a federal judge ruled in September of 2016 that the injury data reported by the authors in the article was false.
It is clear that after that this article cannot be cited. For example, the authors of the meta-analysis excluded this study from their analysis (Claudino JG, 2018).
ANS. Thank you for providing us such important information. We should be more careful when citing the references. We removed the retracted citation, and added some new references.
2.The groups in the study were small, therefore, in statistical analysis, it is mandatory to check the data for normal distribution. When the distribution is different from normal, it is incorrect to use parametric methods of statistics (Student's t-test). Therefore, the conclusions of the study may be incorrect.
ANS. Thank you very much for your suggestion. We have modified our statistical analysis as follows. We first tested the normal distribution of the continuous data (please refer to supplementary table 1). If the distribution was non-normal, we used non-parametric methods of statistics (i.e. Friedman test). If the distribution was normal, we revised the statistical analysis as a two-way repeated measures ANOVA to test the effects of group (HIPT vs TRT) x time (pre vs post).
3.The obtained data are repeated in full in the tables, the text of the manuscript and in the figures. It's redundant. You can recommend the authors to remove the duplication of data, and try to conduct additional data analysis (correlations, add calculated indicators - per kg of body for example)
ANS. Thank you very much for your advice. We combined the figures 3-6 in figure 3 and simplified the description of repeated data in the text to make the paper easier to read. We added the indices of Wingate test per kilogram (i.e. peak power per kg, and mean power per kg) for comparison like the study by Romero-Arenas et al.
4.Table 2 is poorly designed, in particular, there are no units for measuring indicators.
ANS. Thank you for your suggestion. We have added the units for each variable and removed the paired t-test markers.
5.The performance of the Wingate anaerobic test after the High Intensity Power Training program was previously studied by Romero-Arenas S, et al. For some reason, the name of the first author was shortened in the manuscript (Arenas), which is incorrect. It was also necessary to refer to this work (the authors mention the maximum aerobic test in this manuscript, but not about the Wingate anaerobic test) and compare their results with those obtained earlier.
ANS. Thank you very much for your reminder, we have corrected the name. However, Romero-Arenas’s study (2018) also compared the performance of Wingate anaerobic test between the HIPT, TPT and control groups (Table 2), in which improvements have been shown not only in aerobic adaptation (maximum aerobic test), but also in anaerobic capacity. We have revised our discussion. We appreciated for your careful review.
Round 2
Reviewer 1 Report
Overall, the authors addressed all observations and concerns inquired by
I and reviewer 1. Therefore, the quality of the analysis and data are in a good shape for publication.
Author Response
Dear reviewer,
We appreciated for your review and helpful comments that make our manuscript more comprehensive.
Reviewer 2 Report
It must be admitted that the authors tried to take into account the comments as much as possible. However, there is one more important point. Judging by the application, as well as by the methods of non-parametric statistics used, the distribution of some of the quantitative data differed from normal. In such cases, the presentation of quantitative data as mean and standard deviation is incorrect; data should be presented as median and lower and upper quartiles.
Author Response
Dear reviewer,
It must be admitted that the authors tried to take into account the comments as much as possible. However, there is one more important point. Judging by the application, as well as by the methods of non-parametric statistics used, the distribution of some of the quantitative data differed from normal. In such cases, the presentation of quantitative data as mean and standard deviation is incorrect; data should be presented as median and lower and upper quartiles.
Reply: Thank you for your constructive suggestion. However, we used parametric and non-parametric parametric statistics according to the Shapiro-Wilk tests. The assessments that followed normal distribution should be presented as mean and standard deviation (i.e. vertical jump and peak power); otherwise, data should be presented as median and lower and upper quartiles (i.e. bench press and mean power). Accordingly, we re-designed the table 1 & table 2 to provide sufficient information for readers. Thanks again for your careful review that helped our manuscript more comprehensive.